# OpenStamp: A Watermark for Open-Source Language Models

## Abstract

With the growing prevalence of large language model (LLM)-generated content, watermarking is considered a promising approach for attributing text to LLMs and distinguishing it from human-written content. A common class of techniques embeds subtle but detectable signals in generated text by modifying token sampling probabilities. However, such methods are unsuitable for open-source models, where users have white-box access and can easily disable watermarking during inference. Existing watermarking methods that support open-source models often rely on complex or compute-intensive training procedures. In this work, we introduce **OpenStamp**, a simple watermarking technique that implants detectable signals into the generated text by modifying just the final projection, or unembedding, layer. Through experiments across two models, we show that **OpenStamp** achieves superior detection performance, with minimal degradation in model capabilities. The implanted watermarking signal is harder to scrub off through post-hoc fine-tuning compared to previous methods, and offers similar robustness against paraphrasing attacks. We have shared our code through an anonymized repository to enable developers to easily watermark their models.

## 1 Introduction

Large language models (LLMs) are capable of generating human-like text, which has led to their widespread adoption in various applications (Brown et al., 2020; Chowdhery et al., 2023). As these models become more prevalent, there are looming concerns about potential misuse, such as generating large-scale misinformation (Oviedo-Trespalacios et al., 2023), influencing public opinion (Panditharatne & Giansiracusa, 2023), or orchestrating social engineering attacks (Grbic & Dujlovic, 2023). To address these concerns, it is critical to develop methods for distinguishing LLM-generated content from human-written text. Such methods can also be used to attribute the content to its source model, thereby promoting transparency and accountability in LLM use, especially for high-stakes applications. One promising approach to detect model-generated content is to watermark the text by embedding subtle, imperceptible signals into the model's outputs. To implant such signals, several prominent watermarking techniques either modify the next-token probabilities (Kirchenbauer et al., 2023; Liu & Bu, 2024) or constrain the sampling process during generation (Aaronson, 2023; Kuditipudi et al., 2024). While effective, these decoding-based techniques are incompatible with requirements of open sourcing, where users have full control over the generation process and can easily disable the watermarking logic. This motivates the need for techniques that embed watermarking signals directly into the weights of language models.

Few recent works explore this direction for watermarking open-source LLMs. A notable approach proposes to distill a student model using the outputs of the watermarked teacher model (Gu et al., 2024). However, this method typically requires considerable amounts of training data and computational resources. Another recent study proposes to add Gaussian noise to the bias vector of the final layer, steering the generation towards a fixed set of tokens (Christ et al., 2024), also referred to as the green list in watermarking literature. This scheme may have limited applicability as most LLMs omit bias terms in linear layers (Touvron et al., 2023; Jiang et al., 2023; Radford et al., 2019). Furthermore, recent work shows that watermarks relying on a fixed green list can be reverse-engineered and rendered ineffective (Jovanovic et al., 2024; Rastogi & Pruthi, 2024).

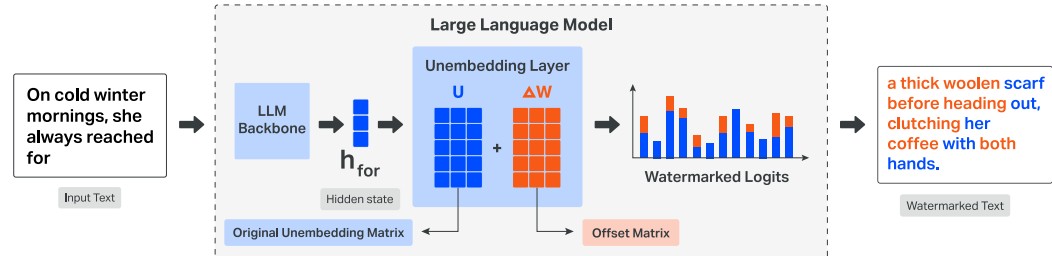

Figure 1: **Overview of our watermarking method.** We add an offset matrix $\Delta W$ to the unembedding layer's weights $U$ to produce **watermark logits** that bias token sampling, favoring tokens with higher watermark logit values. The watermark can be detected using a **log-likelihood ratio (LLR)** based score which measures how much more likely the observed tokens are under the watermarked model compared to the unwatermarked model.

In this work, we introduce `OpenStamp`, a simple approach to watermarking open-source LLMs by modifying the *unembedding layer*[1] weights of the model. We introduce a carefully designed modification to these weights, which bias the logits before generation and thereby implant signals in the generated text. `OpenStamp` is conceptually similar to prior logit-based watermarking approaches (Kirchenbauer et al., 2023; Liu et al., 2024; Liu & Bu, 2024), which influence token selection by adding small, context-dependent biases to the logits at each decoding step. However, an important distinction is that we embed the biasing logic directly into the unembedding layer's weights. Figure 1 shows a high-level overview of our watermarking method. To detect watermarked text, we compute a length normalized **log-likelihood ratio (LLR)** score, which measures how much more likely the observed tokens are under the watermarked model compared to the unwatermarked model.

`OpenStamp` offers multiple advantages over prior work. Unlike past approaches (Gu et al., 2024; Xu et al., 2025; Elhassan et al., 2025), it requires no complex training procedures to embed the watermark. Furthermore, unlike Christ et al. (2024), it does not rely on the final layer bias and cannot be trivially removed. Despite its simplicity, `OpenStamp` achieves near-perfect detection (TPR $\geq 99\%$ at 1% FPR) with only minimal degradation in text quality, outperforming other open-source watermarking methods by *over* 45 *percentage points* in detection accuracy at comparable perplexity levels. Our watermarking method approaches the Pareto frontier defined by prominent decoding-based techniques. Furthermore, our watermarking signal is more difficult to erase via fine-tuning compared to existing approaches and is comparably robust to paraphrasing attacks. We believe that these strengths make our approach a strong candidate for watermarking open-source models. To encourage model developers to watermark their models prior to release, and foster the development of new watermarking techniques for open-source models, we publicly release our code through an anonymized repository: `https://anonymous.4open.science/r/openstamp-78F4/`

## 2 BACKGROUND AND RELATED WORK

**Watermark for Large Language Models.** Watermarking techniques for LLMs embed signals in model-generated text that remain imperceptible to human readers but can be algorithmically detected. A common approach alters the decoding process (Kirchenbauer et al., 2023; Liu & Bu, 2024) by boosting a pseudorandom subset of tokens. Detection relies on statistical tests that exploit the resulting skew, for example by comparing the proportion of green tokens against the baseline expected under unwatermarked text. For a comprehensive survey, see Liang et al. (2024).

**Watermarks for Open-Source LLMs.** In open-source settings, where users control the decoding process, any decoding-time logic can be easily bypassed; hence, the watermarking logic must be embedded directly into the model weights. One approach uses distillation, training a student model on watermarked outputs from a decoding-based watermarked teacher, thereby enabling the student to generate watermarked text natively (Gu et al., 2024). While effective, this method requires substantial computational resources and training data. A reinforcement learning-based framework

---

[1]The unembedding layer is also referred to as the output projection layer or softmax layer.

jointly optimizes an LLM and a paired detector to balance detectability and text quality (Xu et al., 2025). A further line of work embeds a watermark into model weights by jointly fine-tuning LoRA adapters, optimizing for both coherence and detectability (Elhassan et al., 2025). In addition to requiring substantial compute, both methods rely on complex joint optimization objectives that can be challenging to tune.

Similar to our work are approaches that embed watermarking logic without requiring any LLM fine-tuning. One such method introduces a fixed Gaussian perturbation to the final layer's bias vector, steering generation towards positively biased tokens (Christ et al., 2024), similar to a fixed green list (Zhao et al., 2024). However, most LLMs omit bias terms in linear layers (Touvron et al., 2023; Jiang et al., 2023; Radford et al., 2019), and fixed green-list watermarks are known to be vulnerable to reverse-engineering attacks (Jovanovic et al., 2024; Rastogi & Pruthi, 2024). GaussMark (Block et al., 2025) embeds a watermark by adding Gaussian noise to a subset of model weights, and detects it by analysing how the log-likelihood of a given text changes under the perturbed model. However, GaussMark can be difficult to embed, as it requires careful selection of the weight subsets and the noise strength to balance text quality and detectability across models.

Despite recent innovations, a key limitation across existing work is the lack of robustness to post-hoc model modifications. This is a major challenge for watermarking open-source models, which are often updated through quantization, pruning, merging, or fine-tuning. Gloaguen et al. (2025) find that no current methods remain detectable after such updates. We present complementary evidence on this limitation in Section 5.3. Various recent approaches underscore the inherent difficulty of watermarking in open-source LLMs. Each method exhibits different limitations across several axes: efficiency of integration, detectability, robustness, and practicality.

## 3 METHODOLOGY

**Overview.** This section presents our approach for watermarking language model outputs. We first introduce the model setup and notation (§3.1), then describe how a simple modification to the unembedding layer can bias generation towards specific tokens to embed a watermark signal (§3.2). We outline key desiderata for effective watermarking and show how our design satisfies them (§3.3). Finally, we describe how to detect the watermark using a log-likelihood ratio based score (§3.4).

### 3.1 PRELIMINARIES

Let a language model process a sequence of tokens drawn from a vocabulary $\mathcal{V}$. Let $x_t \in \mathcal{V}$ denote the token at generation step $t$, and let $x_{\leq t} = (x_1, \ldots, x_t)$ denote the prefix up to step $t$. The model computes a hidden representation $h_t = f(x_{\leq t}) \in \mathbb{R}^d$, where $f : \mathcal{V}^* \to \mathbb{R}^d$ is the model's internal encoding function. The model then produces a logit vector

$$v_t = U h_t \in \mathbb{R}^{|\mathcal{V}|},$$

where $U \in \mathbb{R}^{|\mathcal{V}| \times d}$ is the unembedding matrix. The model defines a categorical distribution $p(x_{t+1} \mid x_{\leq t})$ over the next token by applying a softmax over $v_t$, where each component $v_t^{(w)}$ corresponds to the logit value of token $w \in \mathcal{V}$. Watermarking strategies typically modify $v_t$ during generation to influence the next-token distribution.

**KGW Watermarking.** In the KGW watermarking scheme (Kirchenbauer et al., 2023), the logit modification step is guided by a pseudorandom partition of the vocabulary. At each generation step $t$, a pseudorandom function (PRF) selects a subset of tokens $\mathcal{G}_t \subset \mathcal{V}$ called the *green list*, based on the context $x_{<t}$ and a secret key. A parameter $\gamma$ controls the fraction of tokens in the green list, so that $|\mathcal{G}_t| = \gamma |\mathcal{V}|$. The logits for tokens in the green list are boosted by a fixed amount $\delta > 0$:

$$\hat{v}_t^{(w)} = v_t^{(w)} + \delta \cdot \mathbf{1}\{w \in G_t\}.$$

### 3.2 WATERMARKING VIA UNEMBEDDING MATRIX MODIFICATION

Modifying the unembedding matrix provides a direct mechanism to alter the logit vector produced at each generation step. Concretely, we define a modified unembedding matrix $\tilde{U} = U + \boldsymbol{\Delta}\mathbf{W}$, where

we refer to $\Delta W \in \mathbb{R}^{|\mathcal{V}| \times d}$ as the *offset matrix*. Applying this matrix to a hidden state $h_t$ yields the modified logit vector $\tilde{v}_t = \tilde{U} h_t = v_t + \mathbf{\Delta W h_t}$, where $\Delta W h_t$ is termed *watermark logits*. These logits bias the output distribution during generation, favoring tokens with higher adjusted scores. This bias accumulates in the generated text, embedding a detectable watermark signal.

Unlike prior approaches that modify arbitrary model weights, altering the unembedding matrix has interpretable effects: the watermark's influence on token probabilities is directly determined by the linear transformation $\Delta W$ applied to hidden states. This linearity allows us to provide a theoretical justification for why the watermark logits remain stable, as discussed in Appendix J.

**Designing an effective offset matrix.** For our method to be effective, the offset matrix $\Delta W$ should embed a watermark that is:

- **Detectable** in generated text using algorithmic methods.
- **Controllable** via hyperparameters that balance detectability and text quality.
- **Variable** across contexts, avoiding fixed biases that can be reverse-engineered.

To meet these desiderata, we now present our design for the offset matrix $\Delta W$.

### 3.3 LINEARIZED GREEN LIST BIASING

Inspired by KGW, where a PRF selects a context-dependent green list $\mathcal{G}_t \subset \mathcal{V}$, we model a similar list-selection mechanism as a composition of two linear transformations. Specifically, we construct the offset matrix as

$$\Delta W = GS,$$

where $G \in \mathbb{R}^{|\mathcal{V}| \times L}$ encodes $L$ green lists, and $S \in \mathbb{R}^{L \times d}$ is designed to map the hidden state $h_t$ to a vector $s = S h_t \in \mathbb{R}^L$ that approximates a one-hot selector for one of the $L$ green lists. The product $Gs \in \mathbb{R}^{|\mathcal{V}|}$ then yields watermark logits that closely resembles one of the encoded green lists in $G$. Because the hidden state encodes the generation context, this construction parallels KGW's use of PRF to tie list selection to context. By maintaining multiple green lists and selecting among them dynamically through $h_t$, the method ensures **variability** in the watermark logits across contexts. We provide supporting evidence of this variability in Appendix G.

**Selector Matrix $S$.** The selector matrix $S \in \mathbb{R}^{L \times d}$ is obtained by solving a ridge regression problem in which input variables are continuous hidden states and the target outputs are one-hot vectors representing $L$ pseudo-classes. To build these pseudo-classes, we first extract the hidden states from the final transformer layer on text sampled from the OpenWebText corpus (Gokaslan & Cohen, 2019). We then group these hidden states into $L$ clusters using the $k$-means algorithm. This gives us a training dataset $\mathcal{D}_{\text{train}} = \{(h_i, e_i)\}_{i=1}^N$, where $e_i$ is the one-hot encoding of the cluster label for $h_i$. We then solve:

$$\min_S \sum_{(h_i, e_i) \in \mathcal{D}_{\text{train}}} \|S h_i - e_i\|^2 + \lambda \|S\|_F^2,$$

where $\lambda > 0$ is a regularization parameter. Once trained, $S$ can be applied to any hidden state $h_t$ to produce a soft class indicator $s = S h_t \approx e_t$. See Appendix B for training details.

**Green List Matrix $G$.** The matrix $G \in \mathbb{R}^{|\mathcal{V}| \times L}$ encodes $L$ candidate green lists as its columns. For each column $l \in \{1, \ldots, L\}$, the corresponding green list $\mathcal{G}_l \subset \mathcal{V}$ is defined by a PRF:

$$\mathcal{G}_l = \{\, i \mid \text{PRF}(\texttt{seed}, l, i) < \gamma \,\},$$

where `seed` is the secret key shared between watermark generation and detection, and $\text{PRF}(\cdot)$ maps the token index $i$ to a value in $[0, 1]$. Each entry of $G$ is then given by

$$G_{i,l} = \delta \cdot \mathbf{1}\{i \in \mathcal{G}_l\}.$$

The hyperparameters $\gamma$ and $\delta$ have the same interpretation as in the KGW watermarking scheme, offering **controllability** over the watermark's strength and impact on text quality.

**Approximating one-hot selectors.** Since our method operates on continuous hidden states, the selector matrix $S$ yields soft, continuous selectors instead of exact one-hot vectors. As a result, multiple green lists can be partially activated at once leading to weaker alignment with the intended green-list behavior. This effect becomes more pronounced as $L$ grows. Nonetheless, detection performance remains strong. A detailed analysis of this effect and its influence on detection performance is presented in Section 5.5.

### 3.4 DETECTION VIA LOG-LIKELIHOOD RATIO

To detect the presence of a watermark, we compute a **length-normalized log-likelihood ratio (LLR)** for the given sequence using the watermarked and original model, specifically:

$$\text{LLR}(x) = \frac{1}{T} \sum_{t=1}^{T} \log \frac{p_{\text{wm}}(x_t \mid x_{<t})}{p_{\text{orig}}(x_t \mid x_{<t})}, \tag{1}$$

where the probabilities are defined via softmax over logits from the original unwatermarked model, $p_{\text{orig}}$ and it's watermarked version, $p_{\text{wm}}$.

**Intuition.** The **LLR** score measures how much more likely a given token sequence is under the watermarked model than under the original model. Because the watermark logits systematically bias sampling towards certain favored tokens, generations from the watermarked model tend to accumulate higher LLR values, making the watermark **detectable** in practice. Length normalization ensures that a single detection threshold can be applied consistently across sequences of different lengths. A key assumption behind this approach is that unwatermarked text, whether human-written or generated by other models, is better modeled by the original model than by the watermarked one. While the conditional probabilities in Equation 1 formally condition on the full history, detection remains robust using only partial prefixes that exclude the generation prompt. We present empirical evidence in Appendix I that detection performance is largely invariant to prompt inclusion.

Unlike frequency-based detection methods (Kirchenbauer et al., 2023), our LLR score does not constitute a formal statistical test. The null hypothesis—that the text is unwatermarked—is *composite*, encompassing human-written and other model-generated text. As noted in Block et al. (2025), this makes p-value calibration intractable, since the null distribution of the LLR is not well-defined. Nevertheless, because the LLR uses the full token probability distribution and naturally captures the *mixture* of green lists produced by our watermarking method, it preserves signal that frequency-based tests lose by relying only on discrete counts. In Section 5.6, we demonstrate this empirically by comparing our LLR-based detector with ablations that discretize the signal.

**Detection Protocol.** The developer publicly releases a watermarked model with the modified unembedding matrix $\tilde{U} = U + \Delta W$, while keeping the original unembedding matrix private. Any text generated using the public model can be detected using Algorithm 1. This protocol assumes white-box access to the model weights in order to evaluate log-likelihoods.

---

**Algorithm 1** Watermark Detection via LLR Score

---

**Require:** Sequence $x = (x_1, \dots, x_T)$, LLM backbone $f(\cdot)$, unembedding matrix $U$, offset matrix $\Delta W$, threshold $\tau$
**Ensure:** `True` if $x$ is watermarked; else `False`
    *// Extract hidden states from the LLM*
1: $(h_1, \dots, h_T) \leftarrow f(x)$
    *// Compute log-likelihood of sequence under original model*
2: $\ell_{\text{orig}} \leftarrow \sum_{t=1}^{T-1} \log \text{softmax}(h_t U^\top)[x_{t+1}]$
    *// Compute log-likelihood under watermarked model*
3: $\ell_{\text{wm}} \leftarrow \sum_{t=1}^{T-1} \log \text{softmax}(h_t (U + \Delta W)^\top)[x_{t+1}]$
    *// Compute length-normalized LLR score*
4: $\text{LLR}(x) \leftarrow (\ell_{\text{wm}} - \ell_{\text{orig}})/(T-1)$
5: **return** $(\text{LLR}(x) > \tau)$

---

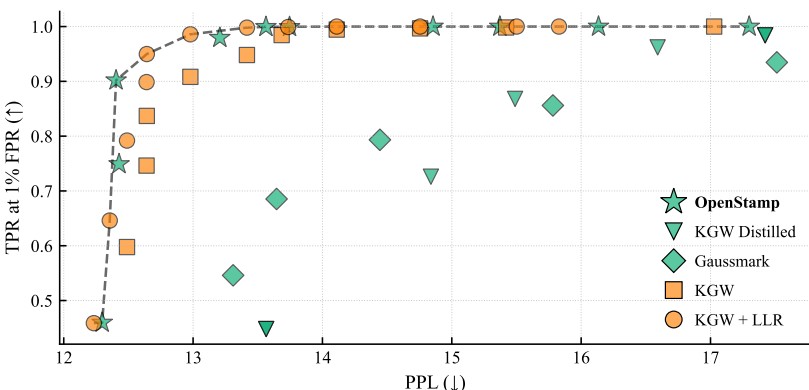

Figure 2: **Trade-off between detectability (TPR at 1% FPR) and text quality (PPL) for various watermarking methods.** KGW + LLR defines the Pareto frontier, establishing an upper bound on detectability at each PPL level. Vanilla KGW performs slightly worse, highlighting the effectiveness of the LLR detection method. Our method, **OpenStamp**, is close to the Pareto frontier and outperforms all open-source compatible methods (shown in green) by a substantial margin.

## 4    EXPERIMENTAL SETUP

We evaluate our watermarking method across four key dimensions: **(i)** detection performance (§ 5.1), **(ii)** robustness to paraphrasing attacks (§ 5.2), **(iii)** resistance to fine-tuning (§ 5.3) and **(iv)** impact on downstream tasks (§ 5.4). We conduct all experiments along these axes on `Llama-2-7B` (Touvron et al., 2023). Additional detection performance experiments are conducted on `Mistral-7B` (Jiang et al., 2023) and presented in Appendix C. Beyond these evaluations, we also provide a deeper analysis of watermarking behavior (§ 5.5) and the LLR detector (§ 5.6).

### 4.1    BASELINES

We compare our method against two prior open-source-compatible watermarking approaches: GaussMark (Block et al., 2025) and a distilled version of KGW (Gu et al., 2024). We also include standard KGW as a representative of more conventional decoding-based techniques. In addition, we evaluate KGW combined with the LLR-based detector described in Section 3.4, which we regard as an approximate upper bound on detection performance. We omit RL-based watermarking (Xu et al., 2025) because it does not remain robust across different sampling strategies (See Appendix E). We omit Elhassan et al. (2025) because we were unable to train a model that could reliably embed a detectable watermark. Hyperparameters for all methods are provided in Appendix D.

### 4.2    EVALUATION PROTOCOL

**Watermarked Sample Generation.**    We generate 500 watermarked completions of 200 tokens each, using 50-token prompts sampled from the `RealNewsLike` subset of C4 (Raffel et al., 2020). The corresponding unwatermarked completions are taken directly from the dataset as the next 200 tokens after each prompt. Detection is performed only on the continuations and excludes the prompts. Watermarked generations are sampled using nucleus sampling with temperature 1.0. To assess generalizability, we also evaluate on prompts drawn from `ArXiv` (Cohan et al., 2018), `BookSum` (Kryściński et al., 2022), and Wikipedia (Wikimedia Foundation, 2024).

**Metrics.**    We evaluate two properties: *watermark detectability* and *text quality*. **Detectability** is measured by the AUROC and the true positive rate at a fixed false positive rate of 1% (TPR@1%FPR). In applications such as plagiarism detection, where false positives carry a high cost, maintaining a low FPR is essential, making TPR@1%FPR a particularly relevant metric. **Text quality** is measured by the mean perplexity (PPL) of the watermarked samples, computed using `Llama-2-13b` as the oracle model. All metrics are averaged over three random seeds, except for KGW Distilled, where only one seed was used due to the high computational cost of distillation.

| Method | ArXiv | | BookSum | | Wikipedia | |
|---|---|---|---|---|---|---|
| | TPR@1%FPR | PPL | TPR@1%FPR | PPL | TPR@1%FPR | PPL |
| **KGW** | 0.99 | 34.1 | 1.00 | 25.1 | 0.97 | 13.7 |
| **KGW + LLR** | 1.00 | 31.9 | 1.00 | 23.5 | 0.99 | 12.5 |
| KGW Distilled | 0.97 | 40.8 | 0.99 | 30.9 | 0.94 | 15.8 |
| GaussMark | 0.92 | 38.6 | 0.95 | 30.1 | 0.90 | 17.0 |
| **OpenStamp** | **1.00** | **31.0** | **1.00** | **26.3** | **0.99** | **14.4** |

Table 1: **Watermark detection performance across datasets.** Prompts are sampled from `ArXiv`, `BookSum`, and `Wikipedia` and evaluated on `LLaMA-2-7B`. Bold values denote the best TPR@1%FPR and lowest PPL per dataset. **KGW and KGW + LLR** are decoding-based methods.

## 5 RESULTS

### 5.1 DETECTION PERFORMANCE

Figure 2 illustrates the tradeoff between text quality and detectability for various watermarking methods. Each method's tradeoff curve is generated by varying a method-specific parameter that controls strength of the signal. Open-source methods are shown in green, while decoding-based methods are shown in orange.

**OpenStamp** achieves near-perfect detection (TPR $\geq$ 99.9%) at a perplexity of approximately 13.6, significantly outperforming Gaussmark and KGW Distilled, which reach only 45-55% TPR at similar PPL levels. The gap between vanilla KGW and the KGW+LLR upper bound highlights the effectiveness of the LLR detection method. As shown in Table 1, our method also outperforms all baselines across other datasets, demonstrating its generalizability. **OpenStamp** shows similar trends across all datasets on `Mistral-7B` (see Appendix C).

We also assess detection under stricter false-positive constraints, since in many applications even a 1% FPR may be too high. Table 2 reports TPR@0.1% FPR and TPR@0.01% FPR for all methods. **OpenStamp** maintains near-perfect detection at both 0.1% and 0.01% FPR, clearly outperforming GaussMark and KGW Distilled.

| Method | TPR@0.1% FPR | TPR@0.01% FPR | PPL |
|---|---|---|---|
| **KGW + LLR** | 1.00 | 1.00 | 14.8 |
| **KGW** | 0.99 | 0.99 | 15.5 |
| Gaussmark | 0.74 | 0.74 | 15.7 |
| KGW Distilled | 0.85 | 0.84 | 16.6 |
| **OpenStamp** | **1.00** | **1.00** | 15.1 |

Table 2: **Detection TPR at stricter FPRs. OpenStamp** maintains near-perfect detection performance even at 0.1% and 0.01% FPR, significantly outperforming GaussMark and KGW Distilled.

### 5.2 ROBUSTNESS TO PARAPHRASING ATTACKS

Paraphrasing attacks pose a major challenge for watermark detection (Krishna et al., 2023; Kirchenbauer et al., 2024). By replacing or rearranging tokens, paraphrasing can dilute the statistical patterns that watermarking methods rely on for detection. To test robustness against such attacks, we use **DIPPER** (Krishna et al., 2023), a paraphrase generation model that allows fine-grained control over *lexical diversity*, a measure of how much the paraphrase's word choice diverges from the original. We apply DIPPER to watermarked samples with two lexical diversity levels, 20 for lighter edits and 60 for stronger edits. As an upper-bound, we include UNIGRAM (Zhao et al., 2024), which uses a static green list and is inherently robust to paraphrasing since its green token set remains fixed regardless of surface form. Note that while UNIGRAM is robust to paraphrasing because of its static green list, it is easier to reverse engineer and thus less secure.

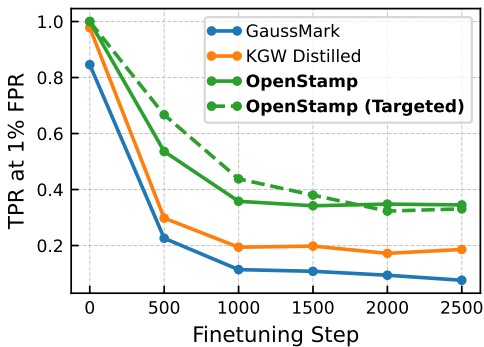 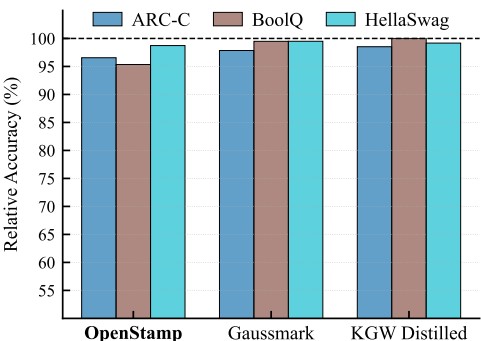

Figure 3: **Detectability under post-hoc fine-tuning.** We report TPR@1%FPR over 2,500 fine-tuning steps. All methods degrade over time, but **OpenStamp** maintains higher detectability compared to GaussMark and KGW Distilled.

Figure 4: **Relative downstream task accuracy of watermarked models.** Accuracy is shown as a percentage of the unwatermarked baseline (dashed line at 100%). All methods exhibit a degradation in accuracy that is less than 5%.

Table 3 shows that **OpenStamp** achieves the highest TPR@1%FPR at low lexical diversity (20), while GaussMark outperforms at higher diversity (60). We observed a similar trend on `Mistral-7B` (see Appendix C). Note that a lexical diversity of 20 corresponds to the kind of quick, low-effort edits a human paraphraser might realistically attempt, making it the more practical setting. In contrast, a diversity of 60 is far less realistic, since achieving that level of rewriting would require effort similar to composing the text from scratch.

| Method | LexDiv = 20 | | LexDiv = 60 | |
|---|---|---|---|---|
| | AUROC | TPR@1%FPR | AUROC | TPR@1%FPR |
| **UNIGRAM** | 0.99 | 0.87 | 0.97 | 0.56 |
| KGW Distilled | 0.97 | 0.71 | 0.87 | 0.29 |
| GaussMark | 0.96 | 0.63 | 0.92 | **0.47** |
| **OpenStamp** | 0.99 | **0.89** | 0.88 | 0.45 |

Table 3: **Paraphrasing attack results on LLaMA-2-7B.** LexDiv (Lexical Diversity) measures the degree of paraphrasing; higher LexDiv means stronger paraphrasing. *UNIGRAM* serves as an upper-bound and is expected to remain robust under paraphrastic transformation. Bold indicates top TPR@1%FPR among methods.

### 5.3 RESISTANCE TO POST-HOC FINE-TUNING

We assess how well open-source compatible watermarking methods can resist post-hoc fine-tuning. Specifically, we simulate an adversary attempting to erase the watermark from model weights by further fine-tuning on OpenWebText (Gokaslan & Cohen, 2019) using LoRA (Hu et al., 2022). Since **OpenStamp** modifies only the unembedding layer, an adversary may leverage this knowledge to perform a more targeted and computationally efficient fine-tuning attack that updates only the unembedding layer. Therefore, we also include this targeted attack in our evaluation. Full experimental details are provided in Appendix F. We measure the TPR@1%FPR after every 500 fine-tuning steps up to 2,500 steps. We discuss more attack setups in Appendix K.

Figure 3 shows that although all watermarking methods degrade over time, **OpenStamp** retains higher detectability than both GaussMark and KGW Distilled. The targeted attack performs similarly to full-model fine-tuning, but serves as a more computationally efficient approach for attempting to erase **OpenStamp**'s watermark.

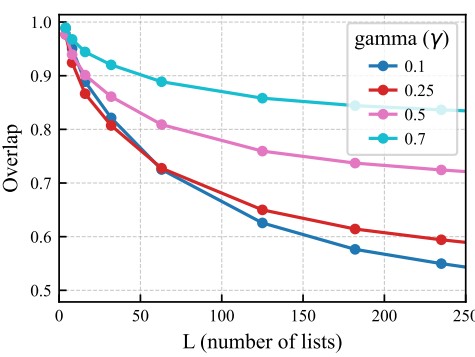 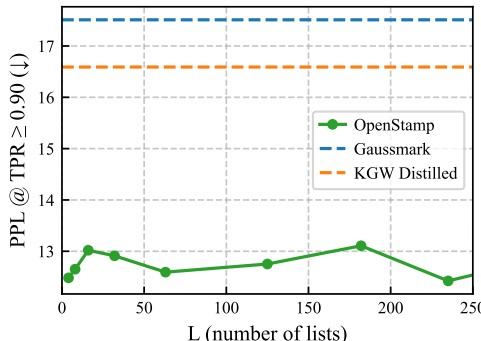

Figure 5: **Green List Token Overlap.** As $L$ increases, the overlap between the top-biased tokens and the intended green list decreases, indicating weakening alignment with the intended green list structure.

Figure 6: **Detection Performance vs.** $L$. Minimum PPL to achieve TPR@1%FPR $\geq 0.90$ is reported for different $L$ values. PPL remains stable with growing $L$ and consistently below the GaussMark and KGW Distilled baselines.

### 5.4 IMPACT ON DOWNSTREAM TASK ACCURACY

Since watermarking strategies alter the output distributions of LLMs, it is essential to ensure that the utility of the underlying model is not significantly compromised. While the results in Figure 2 demonstrate minimal degradation in perplexity, Ajith et al. (2024) find that perplexity measurements cannot reliably predict the performance trade-offs due to watermarking on downstream tasks. Therefore, we further evaluate watermarked models on downstream tasks using the Language Model Evaluation Harness (Gao et al., 2024). We measure the potential performance degradation introduced by watermarking across three benchmarks: ARC-C (Clark et al., 2018), BoolQ (Clark et al., 2019), and HellaSwag (Zellers et al., 2019).

Figure 4 shows the accuracy of various watermarking methods on these tasks, relative to the unwatermarked model. All methods exhibit a degradation in accuracy that is less than 5%, suggesting similar levels of degradation on other downstream tasks.

### 5.5 ANALYSIS OF WATERMARKING BEHAVIOR

In this section, we study how the hyperparameter $L$, which represents the number of green lists, influences our watermarking method. We focus on two aspects: (1) the alignment between watermark logits and the green list structure, and (2) the detection performance.

**Measuring alignment.** We first extract 5,000 hidden states from OpenWebText samples. For each hidden state $h$, we compute watermark logits $\Delta Wh = GSh$. We measure *overlap*, defined as the fraction of $|\mathcal{G}| = \gamma \cdot |\mathcal{V}|$ tokens with the largest logits in $\Delta Wh$ that also belong to the intended green list $\mathcal{G}_\ell$, where $\ell = \arg\max_i(Sh)_i$. We report overlap for different values of $\gamma$ and $L$ in Figure 5. As $L$ increases, the overlap decreases, indicating a weakening alignment with the green list structure.

**Measuring detection performance.** We find the minimum PPL required to achieve TPR@1%FPR $\geq 0.90$ on watermarked samples. A lower PPL threshold indicates a more effective watermark, since it enables reliable detection with less degradation in text quality. We report the PPL threshold for different $L$ values. For comparison, we also report corresponding PPL thresholds for GaussMark and KGW Distilled. Figure 6 shows that PPL remains stable across different $L$ values and consistently below the baseline thresholds.

### 5.6 LLR VS. DISCRETIZED-SIGNAL DETECTORS

We assess how much the LLR detector benefits from retaining the full continuous watermark signal versus *discretizing* it. For this, we compare the LLR detector against two ablations that discretize the watermark signal: (i) a discrete LLR variant that replaces the soft selector with an argmax before

computing likelihoods, and (ii) a binomial count detector that counts the number of selected green list tokens, following the style of Kirchenbauer et al. (2023). In the discrete LLR variant, the selector $s = Sh_t$ is collapsed to a single index $\hat{\ell}_t = \arg\max_i s_i$, and the corresponding green list vector $G_{\hat{\ell}_t} \in \mathbb{R}^{|V|}$ is taken as the discretized signal. The watermarked distribution is then

$$p_{\text{wm}}(x_t \mid G_{\hat{\ell}_t}) = \frac{\exp(v_t + G_{\hat{\ell}_t})_{x_t}}{\sum_{w \in V} \exp(v_t + G_{\hat{\ell}_t})_w},$$

which the mirrors the numerator in Equation 1 but uses only a single green-list vector, removing all mixed-list contributions present in the full model. The binomial count detector instead treats green list membership as a Bernoulli indicator, forming

$$Z = \sum_{t=1}^{T} \mathbf{1}\{x_t \in G_{\hat{\ell}_t}\}, \qquad z = \frac{Z - \gamma T}{\sqrt{T\gamma(1-\gamma)}}.$$

Table 4 shows TPR at various FPR thresholds for all three detectors. The two discretized-signal variants recover some watermark signal but they still perform noticeably worse than the LLR detector, indicating that collapsing the mixed green list structure into a single discrete choice loses information the continuous LLR leverages.

| Method | TPR@1%FPR | TPR@0.1%FPR | TPR@0.001%FPR |
| --- | --- | --- | --- |
| Discrete LLR | 0.96 | 0.82 | 0.79 |
| Binomial Count | 0.86 | 0.51 | 0.50 |
| **LLR** | **1.00** | **1.00** | **1.00** |

Table 4: **Comparing LLR to alternative detectors**. The discretized-signal variants perform worse than the full LLR detector, indicating that collapsing the mixed green list structure into a single discrete choice loses useful information.

## 6 LIMITATIONS

There are several important limitations of our work. First, detection requires a forward pass through the model, making it computationally expensive compared to methods that perform statistical tests on generated text. Second, our method assumes access to token probabilities from the base model. However, this is a realistic assumption as model owners typically have white-box access to their models. While our detection algorithm achieves stronger performance, it is not grounded in a statistical test and thus lacks a probabilistic interpretation, such as confidence levels or false positive rates, that help quantify detection uncertainty. Finally, the performance of our approach, akin to existing work, deteriorates under paraphrasing attacks and post-hoc fine-tuning.

## 7 CONCLUSION

In this work, we proposed a simple and effective approach for watermarking open-source LLMs by embedding the watermarking logic directly into the weights of the unembedding layer. This design enables the watermark to be natively integrated into the models generation process without requiring decoding-time interventions. We demonstrated that our approach outperforms existing open-source watermarking methods in detection performance while preserving text quality. Additionally, we demonstrated that its robustness under paraphrasing and post-hoc model fine-tuning is comparable to, or surpasses, that of existing methods. Overall, our approach offers strong detectability, controllability and can be easily integrated into existing models. Future work could explore more advanced offset matrix designs to improve robustness against paraphrasing and model modifications such as fine-tuning, and to better support the use of detection methods with statistical guarantees.

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

# APPENDICES

## A  EFFECT OF VARYING $\gamma$ AND $\delta$.

We measure the detectability and text quality of watermarked samples generated using different values of $\gamma$ and $\delta$. The results, shown in Figures 7a and 7b, indicate the trade-off between detectability and distortion as we vary these parameters.

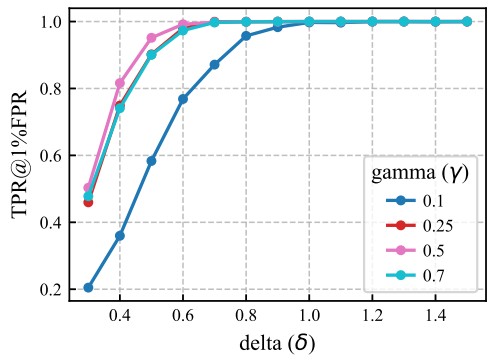
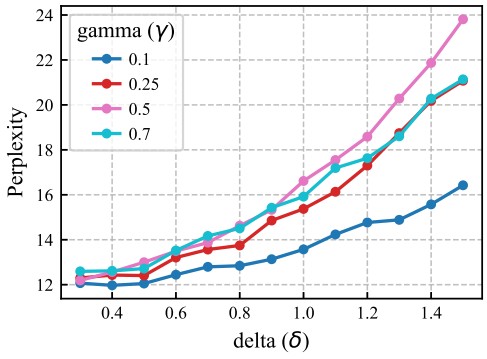

(a) **Effect of $\gamma$ and $\delta$ on detectability.** TPR@1%FPR vs. $\delta$ for various $\gamma$ values.

(b) **Effect of $\gamma$ and $\delta$ on text quality.** Perplexity vs. $\delta$ across the same $\gamma$ settings, measured using `Llama-2-13b`.

## B  TRAINING THE SELECTOR MATRIX

We train the selector projection matrix $S$ to classify hidden state vectors into $L$ distinct classes, where $L$ is an hyperparameter. To obtain the training data, we collect 1.5 million hidden states from the model's final layer by passing OpenWebText (Gokaslan & Cohen, 2019) sequences of maximum length 512 through the model. We apply `MiniBatchKMeans` from `scikit-learn` (Pedregosa et al., 2011) to group these hidden states into $L$ clusters. Clusters with fewer than 10 hidden states are discarded to ensure sufficient representation. We set the ridge regression regularization parameter $\lambda$ to $10^{-3}$ throughout our experiments.

## C  RESULTS ON MISTRAL-7B

We present additional results on the `Mistral-7b` model across two axes: **detection performance** and **robustness to paraphrasing attacks.** We compare our method against GaussMark.

**Hyperparameters.**  For **OpenStamp** , we set the hyperparameters to $\delta = 1.0$, $\gamma = 0.25$, and $L = 203$. For GaussMark, we add noise to the up projection weights of the 20th decoder block with $\sigma = 0.005$.

**Detection Performance.**  We evaluate the detection performance of **OpenStamp** on prompts sampled from the same datasets described in Section 4.2. Table 5 shows that **OpenStamp** achieves a TPR@1%FPR of 1.00 across all datasets, with significantly lower perplexity (PPL) compared to the GaussMark baseline.

**Robustness to Paraphrasing Attacks.**  To evaluate robustness to paraphrasing attacks, we use the same paraphrasing setup as described in Section 5.2. Table 6 shows that **OpenStamp** reports a higher TPR@1%FPR than GaussMark at both levels of lexical diversity.

| Dataset | Metric | GaussMark | OpenStamp |
|---------|--------|-----------|-----------|
| RealNewsLike | TPR@1%FPR | 0.77 | **0.99** |
|  | PPL | 15.3 | **14.1** |
| ArXiv | TPR@1%FPR | 0.78 | **1.00** |
|  | PPL | 34.1 | **26.6** |
| BookSum | TPR@1%FPR | 0.94 | **1.00** |
|  | PPL | 26.9 | **24.6** |
| Wikipedia | TPR@1%FPR | 0.68 | **0.99** |
|  | PPL | 14.4 | **13.1** |

Table 5: **Watermark detection performance for Mistral-7b.** Prompts are sampled from `RealNewsLike`, `ArXiv`, `BookSum`, and `Wikipedia`, evaluated on `Mistral-7B`. Bold values indicate the best TPR@1%FPR and lowest PPL per dataset.

| Method | LexDiv = 20 | | LexDiv = 60 | |
|--------|-------------|---|-------------|---|
|  | **AUROC** | **TPR@1%FPR** | **AUROC** | **TPR@1%FPR** |
| GaussMark | 0.92 | 0.41 | 0.88 | 0.24 |
| **OpenStamp** | 0.98 | **0.78** | 0.87 | **0.30** |

Table 6: **Paraphrasing attack results on Mistral-7B.** Bold indicates top TPR@1%FPR among methods.

# D  HYPERPARAMETER DETAILS FOR WATERMARKING METHODS

This section details the experimental setup and hyperparameters used across all experiments. For each watermarking method, we perform a parameter sweep to generate the plot in Figure 2. For other evaluations, such as robustness to paraphrasing attacks (§ 5.2), downstream task accuracy (§ 5.4), and resistance to post-hoc fine-tuning (§ 5.3), we select a single representative configuration from each method.

**OpenStamp**  We encode $L = 235$ green lists, each containing a fraction $\gamma = 0.25$ of green tokens. The strength parameter $\delta$ is swept over the range $\{0.3, 0.4, \ldots, 1.2\}$ for the Pareto evaluation. For all other experiments, we select $\delta = 1.0$ as the representative configuration.

**GaussMark Baseline**  For GaussMark Block et al. (2025), we perturbed the MLP up-projection weights in a single decoder block using Gaussian noise. For `Llama-2-7b`, the 27th decoder block was used with $\sigma \in \{0.025, 0.03, 0.035, 0.04, 0.045\}$. The configuration $\sigma = 0.04$ is chosen for all other experiments.

**KGW Distilled Baseline**  We trained several logit-distilled KGW variants Gu et al. (2024) using a green list fraction $\gamma = 0.25$ and strength values $\delta \in \{1.0, 1.25, 1.5, 1.75, 2.0\}$ for the Pareto plot. The distilled model with $\delta = 2.0$ was selected for all other evaluations.

**KGW Decoding-Time Watermark**  We used the decoding-time KGW watermark Kirchenbauer et al. (2023), which biases the model's logits during generation without requiring any parameter modification. We set $k = 1$, which is the token context length used by the PRF to generate green lists. We fixed the green list fraction $\gamma = 0.25$ and swept the bias strength $\delta \in \{0.7, 0.8, \ldots, 2.0\}$ to populate the Pareto frontier. For Table 2 and Table 1, we selected $\delta = 1.5$.

**KGW + LLR Detection Variant**  We additionally evaluated KGW with an LLR-based detection strategy (Section 3.4) to simulate a white-box detection setting. The hyperparameters were fixed at $\gamma = 0.25$ and $\delta \in \{0.5, 0.6, \ldots, 1.7\}$ for the Pareto evaluation. For Table 2 and Table 1, we selected $\delta = 1.4$.

## E    PRACTICAL CHALLENGES WITH RL-BASED WATERMARKING

We attempted to implement the RL-based watermarking method from Xu et al. (2025). However, the trained RL model was unable to generate detectable watermarks when the text was generated using multinomial sampling, and it could only embed a strong watermark when sampling with greedy decoding. This is not practical for real-world applications, as greedy decoding often produces repetitive and low-quality text.

We quantify repetition using seq-rep-3, the proportion of duplicate 3-grams in a sequence  (Welleck et al., 2020):

$$1 - \frac{\text{\# of unique 3-grams}}{\text{\# of 3-grams}}$$

We report mean `seq-rep-3` across watermarked samples for both RL watermarking and our method under greedy and multinomial (temperature = 1.0) decoding, along with TPR@1%FPR. Results in table Table 7 show that while greedy decoding makes RL watermarks detectable, it also causes severe repetition, limiting practicality. In contrast, our method preserves low repetition while maintaining strong detectability across both decoding strategies.

| Method | Multinomial | | Greedy | |
|---|---|---|---|---|
| | TPR@1%FPR | seq-rep-3 | TPR@1%FPR | seq-rep-3 |
| RL Watermarking | 0.25 | 0.03 | 0.99 | 0.58 |
| **OpenStamp** | 1.0 | 0.03 | 1.0 | 0.03 |

Table 7: **Detectability (TPR@1%FPR) and text repetition (mean seq-rep-3) for RL watermarking and OpenStamp under multinomial and greedy decoding.** While RL watermarking achieves high detectability with greedy decoding, it causes severe repetition, whereas **OpenStamp** maintains both strong detectability and low repetition across settings.

## F    FINE-TUNING SETUP DETAILS

We fine-tune all models on OpenWebText. We follow a setup similar to  Gloaguen et al. (2025): we use a batch size of 64 with 512 tokens per input and a learning rate of $2e^{-5}$. We use the Adafactor optimizer with cosine learning rate decay and

linear warmup for the first 500 steps. For LoRA Hu et al. (2022), we set the rank to 16, the scaling factor (alpha) to 32, and the dropout rate to 0.1. For our general fine tuning attack we fine-tune all the internal linear layers of the transformer blocks. For the targeted fine-tuning attack on **OpenStamp**, we perform full fine-tuning (i.e., without LoRA) on only the unembedding layer.

## G    EFFECT OF $L$ ON LOGIT VARIABILITY.

We study how increasing $L$ affects the variability of watermark logits across different contexts. With more green lists available, hidden states can be assigned to a greater number of distinct green lists. This increases the diversity of tokens favored by the watermark, making it less predictable and more robust to reverse-engineering attacks. We measure variability by computing the mean Jaccard similarity between the sets of token indices corresponding to the top $\gamma|V|$ components of watermark logits across hidden states. We evaluate this metric for different values of $L$ and $\gamma$. Lower similarity implies greater variability in the watermark logits. As shown in Figure 8, mean similarity decreases with increasing $L$, though the decline plateaus for large $L$, reflecting diminishing marginal gains in variability.

## H    LLM USAGE

ChatGPT (OpenAI, 2025) was used to assist with typesetting (e.g., equations, tables) and for minor language editing (grammar and conciseness). All outputs were reviewed and validated by the authors.

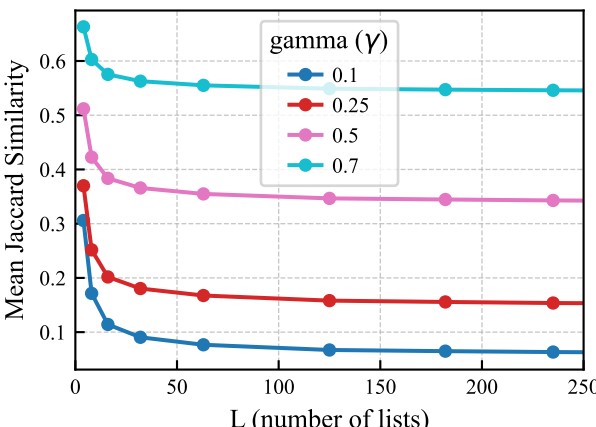

Figure 8: **Variability in Watermark Logits.** As $L$ increases, the mean Jaccard similarity between different top-biased token sets decreases, indicating greater variability in watermark logits across different contexts.

## I  PROMPT INDEPENDENCE OF WATERMARK DETECTION

To explain why our detection method works without the original prompt, we compare per-token LLR scores on watermarked and unwatermarked text, with and without the prompt. As shown in Figures 9a and 9b, the scores are largely consistent across settings, diverging only slightly at the start of generation. This suggests that hidden states—and thus green-list selections—depend mainly on the immediate local context rather than the initial prompt, allowing effective watermark detection even when the prompt is unavailable.

## J  BOUNDEDNESS OF WATERMARK LOGITS

At each step $t$, the watermark logits is defined as $\Delta v_t = \Delta G S h_t$, where $h_t$ is the final-layer hidden state. We assume that hidden states are bounded in norm, which is a reasonable assumption due to the normalization typically applied to final-layer hidden states in Transformer architectures. Furthermore, the selector matrix $S$ is a fixed linear operator with a constant finite operator norm ($\|S\|_{op} = C_S$), and the green list matrix $G$, containing entries in $\{0, \delta\}$, has an operator norm strictly proportional to $\delta$ (i.e., $\|G\|_{op} \leq \delta C_G$). Combining these properties yields a strict upper bound on the perturbation:

$$\|\Delta v_t\|_2 \leq \|G\|_{op}\|S\|_{op}\|h_t\|_2 \leq \delta \cdot (C_G C_S C_h).$$

This implies that the watermark logits are finitely bounded and can be controlled via $\delta$.

## K  ADDITIONAL ATTACK SETUPS

Here we explore several attack setups aimed at erasing the watermark from the weights or recovering the green list assignments.

### K.1  REINITIALIZING AND RETRAINING THE UNEMBEDDING LAYER

An adversary may attempt to completely remove the watermark by reinitializing the unembedding layer and relearning the mapping from hidden states to token logits to restore the model's capability to generate fluent text. However, restoring this capability is non-trivial since model developers train LLMs on large curated datasets and rely on complex pretraining and post-training pipelines.

To illustrate this difficulty, we reinitialized the unembedding layer and retrained only that layer on FineWeb (Penedo et al., 2024). Table 8 reports perplexity after 2,500 and 25,000 training steps.

(a) Watermarked Text

(b) Unwatermarked Text

Figure 9: **Prompt Independence of Watermark Detection.** Per-token LLR scores for (a) watermarked and (b) unwatermarked text, computed with and without the prompt. The red highlight indicates tokens with a negative score whereas the green highlight indicates tokens with a positive score. The text in blue is the prompt. Scores are similar in both settings, indicating that detection is robust to the absence of the original prompt.

Even with substantial retraining, perplexity remained far worse than the original model's, showing that recovering quality is non-trivial and making this attack computationally costly.

| Condition | PPL |
|---|---|
| Watermarked Baseline | 15.1 |
| After 2500 Steps (80M tokens) | 78529.4 |
| After 25000 Steps (800M tokens) | 351.7 |

Table 8: **Perplexity after retraining a reinitialized unembedding layer.**

### K.2    INVERSION ATTACK VIA RECONSTRUCTING THE SELECTOR MATRIX

We evaluate a plausible inversion attack based on the intuition that an adversary might try to reconstruct the selector matrix S and then reverse-engineer each cluster's green list. In practice, this attack is particularly challenging because key details—such as the number of clusters L, the k-means initialization seed, and the dataset used to extract hidden states—are kept private by the model provider.

Even if the attacker guesses L, reproducing the original hidden-state-to-cluster assignments is highly sensitive to both the k-means seed and the underlying dataset. To illustrate this sensitivity, we construct selector matrices with the same $L$ using different datasets (OpenWebText vs. FineWeb) and different initialization seeds. We then generate a test set of hidden states from RealNewsLike samples and compute cluster assignments under each variant of the selector matrix. Agreement between assignments is measured using Adjusted Rand Index (ARI) and Normalized Mutual Information (NMI). As shown in Table 9, cluster assignments vary across datasets and seeds. This variability suggests that reconstructing the original selector matrix is challenging, providing inherent robustness against this class of inversion attacks.

| Condition | ARI | NMI |
|---|---|---|
| Different Dataset | 0.44 | 0.65 |
| Different Seed | 0.46 | 0.67 |
| Different Dataset & Seed | 0.37 | 0.63 |

Table 9: **Agreement between cluster assignments from different selector matrices.** The ARI and NMI scores indicate moderate agreement, showing that reconstructing the original selector matrix is challenging even with knowledge of $L$.

