# OpenReview forum: "OpenStamp: A Watermark for Open-Source Language Models"
_ICLR.cc/2026/Conference — Submitted to ICLR 2026_

### Official Review · Reviewer_3Bfm · 2025-10-28

**Soundness:** 3
**Presentation:** 3
**Contribution:** 3
**Rating:** 6
**Confidence:** 3

**Summary:**

In this paper, the authors propose a watermarking method for open-source language models. The key idea is to embed the watermarking logic directly into the weights of the unembedding layer, allowing the watermark to influence the output distribution without modifying the model architecture. For detection, the authors introduce a length-normalized log-likelihood ratio that compares the likelihoods between the watermarked and base models. The proposed watermark demonstrates strong effectiveness with minimal degradation in model performance, and it remains robust against fine-tuning attacks.

**Strengths:**

- The paper is clear, engaging, and provides well-explained motivations and methods.
- I found the discussion of limitations particularly valuable and appreciate the authors’ transparency in acknowledging them.
- The method is easy to implement and train, and it appears to be readily applicable to a wide range of language models.
- The watermark demonstrates strong robustness against fine-tuning attacks. Notably, even under targeted adaptive attacks, it remains effective after 2,500 fine-tuning steps.

**Weaknesses:**

- The benchmark performance is slightly lower than that of the baseline methods.
- The detection process is relatively expensive.
- As the authors also discussed in the paper, the proposed test is not a formal statistical test, so it does not provide an interpretable metric such as a p-value, unlike some previous works.

**Questions:**

- What happens if the model uses tied word embeddings? Could an attacker simply copy the embedding layer’s weights to completely remove the watermark?
- In the targeted fine-tuning attack, did the authors update the entire unembedding layer, or was LoRA also applied in this case? Can the attacker reinitialize the unembedding layer randomly and do perform the fine-tuning attack?
- Could we use another open-source model (assuming it is not watermarked and shares the same tokenizer) as the base model for testing, so that the detection can be performed by any party?

---

> ### Author Response · Authors · 2025-11-23
>
> We thank the reviewer for their assessment and are happy to see that they appreciate the clarity, practicality, and robustness of our method. We respond to the reviewer’s concerns and questions below.
>
> ### Re: Detection process is relatively expensive
>
> We agree with the reviewer that our test incurs a higher computational cost compared to statistical text-based detection methods, as it requires a forward pass. However, we would like to point out that this characteristic is shared by other open-source watermarking methods [1, 2]. We view reducing detection cost as an interesting direction for future work.
>
> ---
>
> ### Re: Questions
>
> > What happens if the model uses tied word embeddings? Could an attacker simply copy the embedding layer’s weights to completely remove the watermark?
>
> We thank the reviewer for the interesting question. In principle, we agree that if a model uses tied embeddings, an attacker could simply copy the embedding layer’s weights to completely remove the watermark. However, we would like to point out that the majority of recently released open-source LLMs do **not** use tied word embeddings [3,4,5,6], which makes our method widely applicable.
>
> ---
> > In the targeted fine-tuning attack, did the authors update the entire unembedding layer, or was LoRA also applied in this case?
>
> The targeted fine-tuning attack was carried out using LoRA (see Appendix F for details). We also evaluated a more aggressive configuration involving full fine-tuning of the unembedding layer. While we observed a larger decrease in detectability (AUROC = 0.87 after 2,500 steps) compared to the LoRA-based setup  (AUROC = 0.99 after 2,500 steps), our method maintains higher detectability than the baselines. An appendix section has been added detailing the full setup and results of these experiments.
>
> ---
> > Can the attacker reinitialize the unembedding layer randomly and perform the fine-tuning attack?
>
> Yes, randomly reinitializing the unembedding layer would immediately remove the watermark, but it also destroys the model’s ability to generate meaningful text. Restoring that capability is non-trivial, since model developers train LLMs on large curated datasets and rely on complex pretraining and post-training pipelines.
>
> To illustrate this difficulty, we reinitialized the unembedding layer and retrain only that layer on FineWeb [7]. Even after very large amounts of training (> 800 million tokens), the model’s perplexity remained far worse than the original. This demonstrates that recovering quality is not trivial, making this an impractical attack. We have added an appendix section detailing the full setup and results of this retraining attack.
>
> ---
> > Could we use another open-source model (assuming it is not watermarked and shares the same tokenizer) as the base model for testing, so that the detection can be performed by any party?**
>
> We thank the reviewer for suggesting an interesting experiment. Based on the question, we conduct an experiment to find out whether we can use another open-source model for the detection. Specifically, we attempted to detect text generated by a watermarked Llama-2-7B model while substituting the base model in the LLR detector with multiple open-source models.
>
> **Results**
>
> | Base Model         | TPR @ 1% FPR |
> |--------------------|--------------|
> | Llama-2-13B        | **1.00**     |
> | Llama-3.1-8B       | **1.00**     |
> | Phi-4              | 0.79         |
> | Gemma-2-9B         | 0.03         |
> | Qwen-2.5-14B       | 0.87         |
>
> The results indicate that detection works well when the substitute model is similar to the watermarked model.
>
> ---
>
> ### References
>
> [1] Block, A. et al. (2025). **[GaussMark: A Practical Approach for Structural Watermarking of Language Models](https://openreview.net/forum?id=YG3DbpAQBf)**.
>
> [2] Gloaguen, T. et al. (2025). **[Can you Finetune your Binoculars? ](https://arxiv.org/abs/2504.06446)**.
>
> [3] Qwen Team. (2025). **[Qwen3 Technical Report](https://arxiv.org/abs/2505.09388)**.
>
> [4] Groeneveld, D. et al. (2024). **[OLMo: Accelerating the Science of Language Models](https://arxiv.org/abs/2402.00838)**.
>
> [5] AI @ Meta. (2024). **[The Llama 3 Herd of Models](https://arxiv.org/abs/2407.21783)**.
>
> [6] Gemma Team. (2024). **[Gemma 2: Improving Open Language Models at a Practical Size](https://arxiv.org/abs/2408.00118)**.
>
> [7] Penedo, G. et al. (2024). **[The FineWeb Datasets: Decanting the Web for the Finest Text Data at Scale](https://arxiv.org/abs/2406.17557)**.

---

> > ### Comment · Reviewer_3Bfm · 2025-11-25
> >
> > I appreciate the authors’ efforts in addressing my questions, and I intend to maintain my positive score.

---

### Official Review · Reviewer_4SG8 · 2025-10-30

**Soundness:** 3
**Presentation:** 3
**Contribution:** 3
**Rating:** 6
**Confidence:** 4

**Summary:**

This work presents a watermarking method for LLMs that can be open sourced, a known limitation of decoding based schemes that are trivially avoidable in that setting by omitting the decoding scheme when a user runs the model. Their approach embeds an additional transformation within the final lm head/unembedding layer of the model that approximates a soft selection operation over a precomputed set of "greenlists" resulting in a biased token sampling process similar to decoding based watermarks. They evaluate their approach against other open source-able watermarking approaches as well as non-open sourceable decoding based schemes including under paraphrasing and the finetuning attacks more relevant to the open source setting. They demonstrate promising TPR@FPR in both unattacked and attacked settings outperforming the baselines considered.

**Strengths:**

1. Methodology and experimental design are generally clearly presented in S3 and S4; figures are easy to interpret.
2. Evaluation settings are mostly adequate in terms of baselines considered, datasets, and types of ablations on robustness.
3. Method appears to perform competitively and survive attacks (that aren't provided key insider knowledge of the scheme being deployed).

**Weaknesses:**

### Missing evidence for certain design choices

### 1.
Use of a soft selection operation for greenlists makes sense, but has obvious drawbacks. The "fit" process performed is based on a L-way clustering over precomputed hstates to map them to L candidate greenlists, regularized for sparseness. However, the chosen loss is a squared error and the sparseness penalty is the Frobenius norm, an unstructured size minimization penalty. What if this optimization were reworked to more directly minimize the entropy of $Sh_i$? Overall, while a simple choice, which is fine, some evidence for why this particular optim problem is the correct one to showcase the scheme is lacking.

Using cross entropy like $CE(Sh_i,e_i)$ or actually even the reverse $CE(e_i,Sh_i)$ more directly penalize making $Sh_i$ as close to a 1-hot as possible. The reversed version even decomposes into two terms, one of which is precisely the selector entropy one might want to minimize. The decomp referred to is $CE(P,Q) = H(P) + KL(P||Q)$, where canonically P is the ground truth distribution and Q is the predicted or estimated distribution, but the reverse usage is permitted.
It is possible this would improve sample complexity while allowing large numbers of distinct lists to choose from to limit reverse engineering

### 2.
The LLR test appears principled but could the authors explain why the original type of "frequency-based detection method" is not considered as a detection option? L242 motivates that the use of the full logprob ratio test by sensitivity, but the reviewer's intuition is that a better explanation is that the approximate selection via embedded weights method causes predictions to follow mixtures of greenlists not single lists, and thus, it is this mixture that the test is implictly comparing to. Here is an ablation that could help understand the precise way in which the technique is working.

The standard setting is as described in S3.4, however, two alternates are also run. Alternate detector 1) is that p_wm is instead constructed by using \delta W broken into two steps, first applying S to the hstates, then applying an argmax to discretize the selection, then applying the resulting 1-hot to select the greenlist from G, and finally, adding this to the original U but otherwise running Algo 1 the same way. Alternate detector 2) runs the standard setting, the proposed alternate with an argmax in the middle, but simultaneously uses the results of the argmax selection as approximations of a single per token greenlist, and uses this to compute the standard "frequency-based" test. It's not clear precisely how either of these would compare, but the implict expectation based on the author's intuition is that they will be poorer and providing direct evidence for this set of choices in S3.3 and 3.4 would improve the soundness, and interpretability of the proposed method.
(The reviewer notes that this is related to the ablation in Fig 5 which uses argmax overlap, the comment is simply that this same process considered there suggests an important detector performance ablation rather than just an interpretability section at the end)


### Limitations in the robustness evaluation

### 3.

TPR@1%FPR might not be an acceptable threshold in practice as 1/100 error rates are probably too high to deploy and hence it would be helpful if 0.1%FPR or more realistically 0.01%FPR are evaluated. Of course, because the current results and charts use the same target FPR for all methods, the current results are sound, but whether or not each method performs the same at lower values of expected FPR is not clear. One way to show this would be ROC curves highlighting the low FPR region. In that presentation we see whether the open source methods can achieve detection performance at stricter error rates; if ROC curves for different methods ever cross in this region (though they might not) then it indicates that Fig 2 does not reveal the entire story.

### 4.

The experiments in S5.3 regarding the finetuning attack need to be clarified: what distribution is Fig 3 computed over? The referenced App F does not appear to provide the missing details. In these experiments, finetuning is performed on the OWT data for up to 2500 steps, and TPR@FPR is measured throughout. The proposed method conditions the soft watermarking rule implemented by $h_t(U+\Delta W)^T$, on the hstate $h_t$ which is a function of the entire previous prompt context rather than the specified k token window used by schemes like KGW, Aaronson etc. For this reason, it is expected to be more/differently "context dependent" than the n-gram based decoding schemes. Perhaps, the impact of the finetuning attack is localized to just the subdomain of OWT samples considered, so, is  Fig 3 evaluated on those samples? or on the same prompt distributions as in Fig 2 and Table 1?

Also, a note in the writing should be made that the proposed intuition behind the "targeted"-ness of the last layer tuning attack appears to not hold, so it should caveat that immediately at L373 as full finetuning is more effective. This begs the question that there may exist stronger tuning attacks specific to OpenStamp. The point is that iif the experiment had shown major degradation under this notion of targeting, then it would evidence that this targeted attack was indeed properly targeted, but it does not show this. See later comment for a sketch of an alternate targeted attack though of course it is not necessarily a "weakness" to not have considered other options in an initial method proposal paper.

**Questions:**

### 1.
Can the results for the two decoding time baselines included in Fig 2 also be included in Table 1?

### Misc Comments

### 2.
While a reasonable choice of topline number in Table 2, L357 mischaracterizes why "unigram" is more robust than the original KGW. The original method considers context windows for PRF seeding starting at 2 tokens but in followup work considers longer lengths such as 4. The reason for this is that the number of unique PRF seeds, and therefore greenlists is V^k where k is the context window for seeding. This is good for defending against spoofing as the attacker needs to estimate a much larger set of random partitions to learn the watermarking rules from just observed data, however, a wider context means that edits anywhere in that context window change the outcome of the hash and therefore the greenlist selection degrading the watermark. "Unigram" simply selects the most "robust" but simultaneously easiest to "spoof" setting of KGW, and this ease of spoofing is borne out in experiment as at the unigram, context width=0 setting, the watermark is much more rapidly learned than when the width is 1 or 2 tokens; see Figure 2 in Gu et al. 2024, https://arxiv.org/abs/2312.04469.

### 3.
Proposed attack on OpenStamp. Armed with the information that the technique is based on a decomposition between an original $U$ and $\Delta W$ would it be possible to pose this problem directly and learn the linear decomposition? As the size of said matrix is simply the unembedding layer, this is not necessarily an expensive problem to run relative to normal finetuning. Complementarily, what if the adversary simply decides to relearn the unembedding layer (perhaps initialized from the un watermarked embedding layer) by either continuing to train the otherwise frozen model? The point here is that for an open watermarking method to be practical, removing it needs to present a non-trivial computational investment. Since the detection process is not black box, the open source release is expected to include some if not all information about how the watermark test works and so a truly targeted attack will use this information. If the threat model is to assume that only the releaser even knows that the watermark perturbation _exists_, then this is of course a fine academic assumption, but security via obscurity is an undesireable precondition for security that we want to rely on in the wild.

---

> ### Author Response · Authors · 2025-11-24
> **Response to Reviewer 4SG8 (1/2)**
>
> We thank the reviewer for their assessment. The review highlights the clarity of our method and experiments, as well as the competitiveness and robustness of our approach compared to existing open-source watermarking techniques. Below, we address the reviewer's concerns and provide additional results and clarifications.
>
> ---
>
> ## Re: Improving the understanding around why our detectors work
>
> We ran the ablation experiments you proposed to shed light on the efficacy of our detectors. Based on your suggestion, we compared the LLR test to two alternatives:
>
> - **Discrete LLR:** Using an argmax to force a single green list choice before computing the LLR
> - **Binomial Count:** Using that same argmax choice for a simple frequency test similar to the one used in KGW.
>
> | Method         | TPR@1%FPR |
> |----------------|-----------|
> | LLR            | 1         |
> | Discrete LLR   | 0.96      |
> | Binomial Count | 0.86      |
>
> The results show that while both alternatives retain some signal, they are noticeably weaker than our LLR detector. These findings support the reviewer's intuition: because our offset matrix produces a mixture of greenlists, detectors that rely on a discretized choice lose important information. The full LLR detector, by contrast, captures these mixed signals through the full probability distribution, which appears to be critical for achieving reliable detection.
>
> ---
>
> ## Re: Stricter Detection Threshold
>
> We agree that evaluating at stricter FPRs is critical for practical use cases. We ran new tests at 0.1% and 0.01% FPR.
>
> | Method               | TPR@0.1%FPR | TPR@0.01%FPR | PPL  |
> |---------------------|-------------|--------------|------|
> | Openstamp           | 1           | 1            | 15.1 |
> | Gaussmark           | 0.74        | 0.74         | 15.7 |
> | KGW Distilled       | 0.85        | 0.84         | 16.6 |
> | KGW Decoding        | 0.99        | 0.99         | 15.5 |
> | KGW Decoding + LLR  | 1           | 1            | 14.8 |
>
> OpenStamp maintains 100% detection (TPR) even at the 0.01% FPR level, while the other open-source baselines (Gaussmark, KGW Distilled) show a significant drop in performance.
>
> ---
>
> ## Re: Choice of loss function for selector matrix
>
> You’re right in pointing out that ridge regression might not be the most theoretically justified choice for producing one-hot-like vectors. We chose it primarily because its optimal solution has a closed-form analytical solution. This allows us to compute \( S \) in a single, deterministic step, completely avoiding iterative optimization methods like gradient descent.
>
> > Using cross entropy like \( CE(Sh_i, e_i) \) or actually even the reverse \( CE(e_i, Sh_i) \) more directly penalize making \( Sh_i \) as close to a 1-hot as possible.
>
> A standard CE-based loss is incompatible with our core design. It would train \( S \) to produce logits for a non-linear softmax function, forcing our watermark logic to be represented as \( G(\text{softmax}(Sh_t)) \). This non-linear softmax step can't be “baked” into our static, linear \( \Delta W \) matrix. Because our method constrains the selector \( s = Sh_t \) to be purely linear, we're forced to treat it as a regression problem and train \( S \) so that its output directly approximates a one-hot vector (1s and 0s).
>
> ---
>
> ## Re: Attack Setup
>
> > **what distribution is Fig 3 computed over?**
>
> The detection results presented in Figure 3 are computed over the watermarked samples generated using the setup described in Section 4.2 — i.e., over continuations of RealNewsLike prompts.
>
> > **Also, a note in the writing should be made that the proposed intuition behind the "targeted"-ness of the last layer tuning attack appears to not hold, so it should caveat that immediately at L373 as full finetuning is more effective.**
>
> We agree that the “targeted” attack was empirically weaker than full fine-tuning. However, “targetedness” here implies an attacker leveraging white-box knowledge (knowing the watermark is in the unembedding layer) to design a computationally cheaper attack. We wanted to show that an attacker cannot easily scrub the watermark by fine-tuning on the specific layer where it resides. We have updated the manuscript to clarify what we mean by a “targeted” finetuning attack. We also repeat the experiment to do full-finetuning instead of using LoRA to present a stronger targeted attack.
>
> | Setup                               | TPR @ 1% FPR after 2500 steps |
> |-------------------------------------|-------------------------------|
> | Openstamp — All Layers (LoRA)        | 0.34 |
> | Gaussmark — All Layers (LoRA)        | 0.07 |
> | KGW Distilled — All Layers (LoRA)    | 0.18 |
> | **Openstamp — Unembedding Layer (Full)** | 0.30 |
>
> The results show that while targeted fine-tuning on Openstamp leads to degradation, it still maintains higher detectability compared to the baselines.

---

> > ### Author Response · Authors · 2025-11-24
> > **Response to Reviewer 4SG8 (2/2)**
> >
> > ## Re: Questions
> >
> > ### **Question 1:**
> > > Can the results for the two decoding time baselines included in Fig 2 also be included in Table 1?
> >
> > We have added the results for both decoding-time baselines to Table 1 in the revised manuscript. Consistent with the trends in Figure 2, we find that KGW performs comparably to OpenStamp on these datasets, while KGW + LLR achieves the strongest performance overall.
> >
> > ---
> >
> > ### **Question 2:**
> > > L357 mischaracterizes why "unigram" is more robust than the original KGW.
> >
> > While we employed Unigram solely as an empirical upper bound for robustness in our paraphrasing experiments, we acknowledge that Unigram essentially operates as the context-independent case (\( k=0 \)) of the KGW scheme, trading spoofing resistance for maximised robustness against lexical edits. We have revised Section 5.2 to explicitly clarify that Unigram’s robustness stems from its context independence, and we have noted the associated vulnerability to spoofing as established in Gu et al. (2024). [1]
> >
> > ---
> >
> > ### **Question 3:**
> > > Armed with the information that the technique is based on a decomposition between an original \( U \) and \( \Delta W \) would it be possible to pose this problem directly and learn the linear decomposition?
> >
> > To the best of our knowledge, an adversary attempting to learn the decomposition would need to reconstruct the selector matrix $S$ to estimate the specific green list assignments for each cluster. Rebuilding $S$ is difficult, as it requires knowing the exact number of clusters $L$, the k-means initialization seed, and the specific dataset used to extract hidden states. Even if the attacker guesses $L$, reproducing the original hidden-state–to–cluster assignments is highly sensitive to both the k-means seed and the underlying dataset. To demonstrate this, we construct selector matrices using different datasets (OpenWebText vs. FineWeb [2]) and different seeds, then measure clustering agreement on a shared test set using Adjusted Rand Index (ARI) and Normalized Mutual Information (NMI). The average scores (ARI $\approx$ 0.4, NMI $\approx$ 0.65) show disagreement, indicating that the true clustering cannot be reliably recovered. This suggests that $\delta W$ decomposition cannot learned easily. The full setup and results of this attack are detailed in Appendix K.
> >
> > > What if the adversary simply decides to relearn the unembedding layer (perhaps initialized from the unwatermarked embedding layer) by either continuing to train the otherwise frozen model?
> >
> > Randomly reinitializing the unembedding layer destroy the model's capability to generate meaningful text. Relearing this capability is non-trivial, since model developers train LLMs on large curated datasets and rely on complex pretraining and post-training pipelines.
> >
> > To illustrate this difficulty, we reinitialized the unembedding layer and retrain only that layer on FineWeb. Even after significant training (> 800 million tokens), the model’s perplexity remained far worse than the original. This demonstrates that recovering quality is not trivial, making this an impractical attack. The full setup and results of this retraining attack are detailed in Appendix K.
> >
> > | Condition               | PPL  |
> > |-------------------------|------|
> > | Watermarked Baseline        | 15.1 |
> > | After 2500 Training Steps (80M tokens)   |  78529.4    |
> > | Ater 25000 Training Steps (800M tokens)   |  351.7 |
> >
> > ---
> >
> > We hope this response has addressed your concerns and would request you to kindly reconsider your overall assessment. Please let us know if we can address any further concerns.
> >
> > [1] Gu, C. et al. (2024). **[On the Learnability of Watermarks for Language Models](https://openreview.net/forum?id=9k0krNzvlV)**.
> >
> > [2] Penedo, G. et al. (2024). **[The FineWeb Datasets: Decanting the Web for the Finest Text Data at Scale](https://arxiv.org/abs/2406.17557)**

---

> > > ### Comment · Reviewer_4SG8 · 2025-11-26
> > > **Authors' rebuttal response clarifies and strengthens work.**
> > >
> > > I appreciate the authors' work in responding to the points raised in my review and their willingness to explore additional analyses proposed. While I am of course biased, the clarifications and results presented in their response and the updated manuscript improve the quality and completeness of the work. So I will be happy to more fully recommend acceptance conditioned on a few final updates to the draft.
> > >
> > > ---
> > >
> > > Cont'd discussion to selected points only; I have read all parts of the response and perused the updated draft:
> > >
> > > 1.  Even if it feels like a superfluous experiment to include, I think that the frequency test versions in contrast to the LLR test are valuable to include for both soundness and clarity. I would mention this at L250-L256 since you note in that paragraph that the LLR test doesn't admit a simple null distribution and so this detector sacrifices the ability to write down a simple statistical test. What is clear is that your empirical results suggest that the tradeoff is worthwhile, but including this comment and experiment will make that very clear at the intuitive moment in the paper's story when the design choice is being made.
> > >
> > > 2. It greatly strengthens the potential contribution of the paper if the results at 0.1% and 0.01% FPR are included in the manuscript. The results suggest that not only does OpenStamp handily outperform the considered open-sourceable baselines but that it also recovers the power of the decoding time method from which it is derived at stricter error tolerances. This is a very strong empirical result that not only just makes the method "look" better but is more likely to actually encourage further testing and usage by practitioners.
> > >
> > > 3. Thank you for the clarification. Apologies for the misunderstanding regarding the linearizability requirements of the design; this choice does make sense.
> > >
> > > 4. The additional attack discussions and ablations are appreciated. On the finetuning approaches, the LoRA versus Full ablation is probably important to include in the draft whereas the full last layer deletion and recovery is perhaps less informative. As the cluster discovery was discussed in more than one reviewer discussion, it is also potentially important to include; though the author suspects that this threat model might yield stronger attacks than the one considered. Overall, in my opinion, a method proposal paper like this one is required to at least discuss potential attacks, and is of course greatly improved by actually executing some representative attacks. However, the onus is not fully on the initial paper to necessarily discover the most powerful attack and so the collection presented seems adequate even if only to prompt motivated attack papers to follow it up :']
> > >
> > > PS... Was Eq. 1 meant to be changed in some way? The red and blue looked the same to me while skimming.
> > >
> > > ---
> > >
> > > If the authors generally agree with the suggestions above and can update the draft to include a few more of the key improved results, I will be happy to increase my score.

---

> > > > ### Author Response · Authors · 2025-11-28
> > > > **Adding changes to the manuscript as requested by the reviewer**
> > > >
> > > > Thank you for the constructive follow-up. We’ve incorporated the suggested updates into the revised draft (changes marked in blue). A brief summary of the updates is below:
> > > >
> > > > > Even if it feels like a superfluous experiment to include, I think that the frequency test versions in contrast to the LLR test are valuable to include for both soundness and clarity
> > > >
> > > > We have added the alternate-detector comparison in **Section 5.6**.
> > > >
> > > > > It greatly strengthens the potential contribution of the paper if the results at 0.1% and 0.01% FPR are included in the manuscript.
> > > >
> > > > We now include these stricter-FPR results in **Table 2**.
> > > >
> > > > > On the finetuning approaches, the LoRA versus Full ablation is probably important to include in the draft
> > > >
> > > > **Figure 3** has been updated to include the targeted full-finetuning attack on the unembedding layer; the remaining attack setups (retraining head from scratch, selector reconstruction) are kept in the appendix due to the 10-page limit.
> > > >
> > > > > PS... Was Eq. 1 meant to be changed in some way? The red and blue looked the same to me while skimming.
> > > >
> > > > This was only a formatting change to add a numbered reference; `latexdiff` highlighted it unnecessarily. We’ve removed the highlight to avoid confusion.
> > > >
> > > > We appreciate the clear guidance. These additions substantially strengthen the draft. We’re also happy to continue the discussion or make any further adjustments that would be helpful.

---

### Official Review · Reviewer_Uqzi · 2025-10-31

**Soundness:** 2
**Presentation:** 3
**Contribution:** 2
**Rating:** 4
**Confidence:** 5

**Summary:**

This paper proposes OpenStamp, a weighted language model watermarking method designed to embed detectable watermark signals into open-source large language models. Unlike previous decoding-based watermarking approaches that modify the decoding logic during inference, OpenStamp directly integrates the watermarking mechanism into the model’s weights by altering only the final unembedding layer. Specifically, the authors introduce a Linearized Green-List Biasing mechanism, which uses a low-rank matrix factorization $\Delta W = G S$ to approximate the PRF-based random green-list selection process in KGW. Here, $G$ stores $L$ fixed green lists (each containing approximately $\gamma \cdot |V|$ tokens), while $S$ maps the hidden state $h_t$ to a near one-hot selection vector. During generation, the model’s output logits are modified as $v_t' = (U + \Delta W) h_t$ where the term $\Delta W h_t$ serves as a context-dependent bias. For detection, the authors propose a statistical test based on the length-normalized log-likelihood ratio (LLR), which compares the conditional probability of the text under the watermarked and original models to distinguish watermarked outputs from unwatermarked ones.

**Strengths:**

The paper presents a clear and well-structured idea of embedding decoding-based watermarking behavior directly into model weights.

**Weaknesses:**

1. The application scenario is confusing. In Section 3.4, the proposed LLR-based detection requires computing per-token conditional probabilities $p(x_t|x_{<t})$, which implies that the full prompt of the generated text must be known. If the prompt is unavailable (e.g., only the continuation text or human-written text is given), detection cannot be performed. Therefore, this method is more suitable for ownership verification scenarios, where the model owner audits its own generations, rather than for general content detection for open-source LLMs. If the intended goal is to distinguish between human-written and LLM-generated text, it is unclear how one can obtain the prompt for an arbitrary text (or how ``prompt'' applies to human text). The paper should clearly define its target use case, as the current presentation makes the intended scenario ambiguous. If it is used for ownership verification of model, what is the difference between it and the backdoor techniques for model watermarking?

2. The attack model is weak. The robustness evaluation only considers mild parameter updates (finetune the last ten decoder layers together with the unembedding layer). In realistic and academic adversarial settings, attackers are far more capable. For example, distilling the watermarked model into a new student model, or retraining or replacing the output head $U$, directly removing the $\Delta W$ bias. These operations are computationally affordable and potentially effective for a professional attacker. Hence, the paper's claim of being ``robust to fine-tuning'' only holds under a narrow and weak threat model.

3. OpenStamp’s imitation of a key is extremely limited. In my mind, I understand L as the number of vocabulary permutations in KGW's watermark algorithm. However, in KGW, the number of vocabulary permutations is $|V|!$. OpenStamp predefines only $L=235$ lists, far smaller than the vocabulary ($|V|\approx32, 000$ in llama models). The small $L$ reduces dynamic variability.  Because $S h_t$ is not perfectly one-hot, multiple green lists can be softly activated at once, diffusing bias energy across many tokens, which increases PPL (in Table 3, RealNewslike PPL$\approx$14). Since $S h_t$ is a linear and learnable mapping, an attacker with access to $(h_t, logits)$ pairs could reconstruct the pattern of $G$ or the seed-dependent structure. Therefore, OpenStamp ``mimics'' a key but lacks its cryptographic protection. In decoding-based watermarking, the seed can remain private (embedded in the decoder only). In OpenStamp, the seed is fixed during training and embedded into $G \rightarrow \Delta W \rightarrow$ model weights. Therefore, for open-source models, this key is fully exposed, making the watermark both visible and reversible. An attacker could estimate the $G$ pattern and perform a reverse or spoofing attack [1].

[1] Watermark Stealing in Large Language Models, Nikola Jovanović, Robin Staab, Martin Vechev, ICML 2024

**Questions:**

1. What is the application scenario?

2. What is the perplexity of nonwatermarked text?

3. How is its resilience against spoofing attacks?

---

> ### Author Response · Authors · 2025-11-24
> **Response to Reviewer Uqzi (1/2)**
>
> We thank the reviewer for their assessment. The review highlights the clarity and structure of the idea behind our method. It also raises some important concerns that we address below.
>
> ### Re: Clarifying Prompt Access and Intended Application Scenario
>
> We sincerely thank the reviewer for raising this point regarding Section 3.4. We understand how the expression for the conditional probability $(p(x_t \mid x_{<t})$ might be interpreted as requiring access to the full input prompt during detection.
>
> To clarify, **our detection method does not require access to the prompt**. All experiments in the paper are performed solely on generated continuations **without** using the original prompt (as noted in L313). Although the conditional probability formally depends on the entire prefix, in practice the method functions effectively with only the available local context. This is because the hidden states (and the resulting green list selections) are primarily influenced by the immediate context rather than the initial prompt.
>
> To prevent any further confusion, we have updated Section 3.4 to explicitly state that prompt access is *not* required for detection. Additionally, we have added Appendix I, which provides empirical evidence of prompt independence, analysing the stability of the detection method with and without the prompt.
>
> We hope this clarification resolves the reviewer’s concern regarding the intended application scenario. Our method is designed for **watermarked text detection in the wild** i.e., verifying whether text was produced by the released, watermarked model.
>
>
> ### Re: Weak attack model
>
> > The robustness evaluation only considers mild parameter updates (finetune the last ten decoder layers together with the unembedding layer)
>
> Thank you for raising this concern. We re-ran the experiments in Section 5.3 utilizing LoRA on all layers for the general attack and full fine-tuning (instead of LoRA) on the unembedding layer for the targeted attack.
>
> The results (see table below) demonstrate that even under these aggressive parameter updates, Openstamp  outperforms baselines like Gaussmark and KGW Distilled. We have updated the manuscript with these findings.
>
> | Setup                               | TPR @ 1% FPR after 2500 steps |
> |-------------------------------------|-------------------------------|
> | Openstamp — All Layers (LoRA)        | 0.34 |
> | Gaussmark — All Layers (LoRA)        | 0.07 |
> | KGW Distilled — All Layers (LoRA)    | 0.18 |
> | Openstamp — Unembedding Layer (Full) | 0.30 |
>
> > In realistic and academic adversarial settings, attackers are far more capable. For example, distilling the watermarked model into a new student model, or retraining or replacing the output head $U$, directly removing the $\Delta W$ bias. These operations are computationally affordable and potentially effective for a professional attacker.
>
> We thank the reviewer for raising this important point. To address the concern regarding the "retraining or replacing the output head" attack, we perform the following experiment:
>
> **Retrain unembedding layer:** We re-initalize the watermarked unembedding layer and finetune it to restore it's capability to generate coherent text. Restoring this capability is non-trivial, since model developers train LLMs on large curated datasets and rely on complex pretraining and post-training pipelines. To illustrate this difficulty, we reinitialized the unembedding layer and retrain only that layer on FineWeb [1]. Even after training on 800 million tokens, the model fails to recover original performance.
>
> | Condition               | PPL  |
> |-------------------------|------|
> | Watermarked Baseline        | 15.1 |
> | After 2500 Training Steps (80M tokens)    |  78529.4    |
> | Ater 25000 Training Steps (800M tokens)   |  351.7 |
>
> The setup details and results for this experiment are provided in Appendix K. We are open to performing additional experiments if the reviewer can provide references or implementation details to other attack strategies.

---

> > ### Author Response · Authors · 2025-11-24
> > **Response to Reviewer Uqzi (2/2)**
> >
> > ### Re: Limited variability
> > > OpenStamp’s imitation of a key is extremely limited. In my mind, I understand L as the number of vocabulary permutations in KGW's watermark algorithm. However, in KGW, the number of vocabulary permutations is $|V|!$. OpenStamp predefines only $L=235$ lists, far smaller than the vocabulary.
> >
> > Thank you for the comment. While KGW has a theoretical permutation space of \(|V|!\), the relevant quantity for watermark unpredictability is the **effective variability realized during generation**. Appendix G analyzes how \(L\) influences variability within OpenStamp; to directly address your concern, we additionally compared OpenStamp against KGW using the same variability metric.
> >
> > We first compute the green-token sets across 5000 RealNewsLike prefixes for both methods. For Openstamp, we define the green-token set for a prefix as the top $\gamma|V|$ tokens under the watermark logits i.e. the subset of tokens the watermark boosts most strongly for that prefix.
> > We then measure variability via the mean inter-pair Jaccard similarity of these green-token sets (lower = more variability):
> >
> > | Method                   | Mean Jaccard ↓ |
> > |--------------------------|----------------|
> > | KGW (k = 1)              | 0.147          |
> > | KGW (k = 2)              | 0.143          |
> > | OpenStamp                | 0.155          |
> >
> > These values are nearly identical, showing that **OpenStamp achieves practical variability on par with KGW**, even with a finite number of lists.
> >
> > > Since $S h_t$ is a linear and learnable mapping, an attacker with access to $(h_t, logits)$ pairs could reconstruct the pattern of $G$ or the seed-dependent structure.
> >
> > Thank you for raising this concern. While \(S h_t\) is a linear and learnable mapping, recovering the underlying green-list structure from \((h_t, \text{logits})\) pairs is not trivial. Reconstructing \(S\) requires replicating the exact clustering used during watermark construction. However, as per our assumption, the parameters that determine this clustering — the number of clusters \(L\), the k-means initialization seed, and the dataset used to sample hidden states — are all private. Even if the attacker has access to L, reproducing the original hidden-state–to–cluster assignments is highly sensitive to both the k-means seed and the underlying dataset. To demonstrate this, we construct selector matrices using different datasets (OpenWebText vs. FineWeb) and different seeds, then measure clustering agreement on a shared test set using Adjusted Rand Index (ARI) and Normalized Mutual Information (NMI). The average scores (ARI ≈ 0.43, NMI ≈ 0.65) show disagreement, indicating that the true clustering cannot be reliably recovered.
> >
> > We hope our responses help alleviate the reviewer's concerns about spoofing attacks. We are open to performing additional experiments if the reviewer can provide references or implementation details to other attack strategies.
> >
> > ### Re: Questions
> >
> > > What is the perplexity of nonwatermarked text?
> >
> > The mean perplexity of human samples from the realnewslike dataset, computed using a LLaMA-13B is **6.1**. The mean perplexity for unwatermarked model continuations of the realnewslike prompts is **12.1 for Llama-2-7B**  and **11.6 for Mistral-7B**.
> >
> > ---
> > We hope this response has addressed your concerns and would request you to kindly reconsider your overall assessment. Please let us know if we can address any further concerns.
> >
> > [1] Penedo, G. et al. (2024). **[The FineWeb Datasets: Decanting the Web for the Finest Text Data at Scale](https://arxiv.org/abs/2406.17557)**

---

### Official Review · Reviewer_2c5R · 2025-11-01

**Soundness:** 3
**Presentation:** 3
**Contribution:** 2
**Rating:** 4
**Confidence:** 4

**Summary:**

This paper introduces OpenStamp, a watermarking technique for open-source LLMs that modifies the unembedding (output projection) layer to embed a detectable signal in generated text. Detection uses a log-likelihood ratio test comparing the watermarked and base models. Unlike decoding-time watermarking, this method remains active even when users modify decoding, making it suitable for open-source release scenarios. Experiments on LLaMA-2-7B and Mistral-7B show strong detection performance, minimal perplexity increase, and resistance to post-hoc fine-tuning and paraphrasing attacks.

**Strengths:**

- Practical, simple idea: modifying the final projection layer is lightweight and training-free, lowering barriers to adoption.

- Strong empirical performance: consistently high TPR at 1% FPR across datasets, outperforming baselines in most settings.

- Relevant problem setting: watermarking for open-source models is increasingly important as white-box access grows.

**Weaknesses:**

- Limited theory: The paper does not provide a deeper analysis of why the perturbation remains stable across long-range generation or how the perturbations interact with model dynamics.

- Model scale: All experiments are on ~7B models. Performance and safety tradeoffs at larger scales (e.g., 34B/70B) are not explored, limiting confidence in generalization.

- Paraphrasing robustness at high lexical diversity: While performance at LexDiv=20 is strong, the drop at LexDiv=60 suggests the approach remains vulnerable under aggressive rewriting.

- Attack model coverage: Beyond simple fine-tuning, other relevant attacks (e.g., structured pruning, low-rank weight replacement, extraction/inversion attacks) are not evaluated.

- Incremental novelty: Conceptually close to existing logit-bias watermarking approaches; contribution is largely engineering efficiency + practical insight, rather than algorithmic novelty.

Overall, the work is practically relevant, but feels one step short of readiness for a top-tier venue, primarily due to limited evaluation scope.

**Questions:**

- How does accuracy degrade for larger models? Have you tested 13B/34B or 70B variants?

- Are there conditions on model architecture or token distribution under which the watermark signal may fail to propagate?

- How resilient is the watermark to common deployment transformations such as quantization, pruning, or weight merging

---

> ### Author Response · Authors · 2025-11-24
> **Response to Reviewer 2c5R (1/2)**
>
> We thank the reviewer for their feedback. We appreciate their recognition that our method is simple, easy to adopt, and shows strong empirical performance. Below, we address their concerns and provide additional clarifications and results.
>
> ### Re: Limited Theory
> > The paper does not provide a deeper analysis of why the perturbation remains stable across long-range generation or how the perturbations interact with model dynamics.
>
> We appreciate the request for deeper theoretical grounding of the stability of logit perturbations. In response, we have added Appendix J, which provides a formal upper bound on the watermark perturbation. Using the fact that Transformer normalization layers keeps $h_t$ within a stable norm range, we show that the induced logit perturbation $\Delta \|v_t\|_2$ is strictly bounded and scales linearly with $\delta$, ensuring stability throughout long-form generation.
>
> ---
> ### Re: Model Scale Generalizability
> > Performance and safety tradeoffs at larger scales (e.g., 34B/70B) are not explored, limiting confidence in generalization.
>
> We evaluated our method on two larger LLMs, Qwen2.5-32B and Llama-3.1-70B, using the setup described in Section 4.3. The table below reports the detectability (TPR@1%FPR). We also include results for the 7B models from our initial experiments for comparison. The results show that our method is effective at larger scales.
>
> | Model               | TPR@1%FPR | PPL  |
> |---------------------|-----------|------|
> | Llama-2-7B          | 1.00      | 15.1 |
> | Mistral-7B-v0.3     | 0.99      | 14.1 |
> | **Qwen2.5-32B**         | 0.99      | 10.9 |
> | **Llama-3.1-70B**       | 0.98      | 13.7 |
>
> ---
> ### Re: Incremental novelty
> >  Conceptually close to existing logit-bias watermarking approaches;
>
> We respectfully disagree with the characterization of our work as having only incremental novelty. While it is true that our method is inspired by logit-bias watermarking approaches, we believe that baking the watermarking logic directly into the model weights represents a non-trivial contribution that goes beyond simple engineering efficiency.
>
> ---
> ### Re: Paraphrasing Robustness
> We acknowledge that this is a limitation of our method, as well as of other watermarking approaches. That said, our method performs well at lower lexical-diversity levels, which correspond to the more common cases of light editing. We leave improving robustness to more aggressive rewriting as future work.
>
> ---
> ## Re: Questions
> > Have you tested 13B/34B or 70B variants?
>
> We tested our method on Qwen2.5-32B and Llama-3.1-70B. The results are presented in **Re: Model Scale Generalizability**.
>
> ---
> > Are there conditions on model architecture or token distribution under which the watermark signal may fail to propagate?
>
> Like other watermarking methods, we suspect our method would struggle with low-entropy texts where the distribution of valid next tokens is highly constrained. To the best of our knowledge, there are no architectural conditions that would prevent signal propagation, assuming the model has a standard LLM architecture.
>
> ---
> > How resilient is the watermark to common deployment transformations such as quantization, pruning, or weight merging?
>
> We present preliminary results to understand the resilience of our watermark against quantization. We apply inexpensive, popular zero-shot quantization methods on our watermarked models and evaluate detection performance.
> | Quantization | TPR@1%FPR |
> |---|---|
> | INT8 (8 bits) | 1.00 |
> | NF4 (8 bits) | 1.00 |
>
> Our results demonstrate **no drop in performance** in accordance with prior work that quantization does not seem to affect weight based watermarks. [1]

---

> > ### Author Response · Authors · 2025-11-24
> > **Response to Reviewer 2c5R (2/2)**
> >
> > ### Re: Attack Model Coverage
> > Thank you for highlighting the limitations in our attack model coverage. In response to your feedback, we have evaluated our method against additional attack setups. We also repeated the experiments in Section 5.3 with a more aggressive fine-tuning configuration, updating the results accordingly in the revised manuscript.
> >
> >
> > **Targeted fine-tuning on unembedding layer:**
> > We perform targeted fine-tuning directly on the unembedding layer using full fine-tuning.
> >
> > | Setup                               | TPR @ 1% FPR after 2500 steps |
> > |-------------------------------------|-------------------------------|
> > | Openstamp — All Layers (LoRA)        | 0.34 |
> > | Gaussmark — All Layers (LoRA)        | 0.07 |
> > | KGW Distilled — All Layers (LoRA)    | 0.18 |
> > | **Openstamp — Unembedding Layer (Full)** | 0.30 |
> >
> > The results show that while targeted fine-tuning on OpenStamp leads to degradation, it still maintains higher detectability compared to the baselines.
> >
> > **Retrain unembedding layer:** We re-initalize the watermarked unembedding layer and finetune it to restore it's capability to generate coherent text. Restoring this capability is non-trivial, since model developers train LLMs on large curated datasets and rely on complex pretraining and post-training pipelines. To illustrate this difficulty, we reinitialized the unembedding layer and retrain only that layer on FineWeb [2]. Even after training on 800 million tokens, the model fails to recover original performance.
> >
> > | Condition               | PPL  |
> > |-------------------------|------|
> > | Watermarked Baseline        | 15.1 |
> > | After 2500 Training Steps    |  78529.4    |
> > | Ater 25000 Training Steps    |  351.7 |
> >
> >
> >
> > **Low-rank Weight Replacement:** We replace the final layer with a low-rank approximation obtained via SVD. We found that even with a rank of 1024, the LLM generates incoherent text (PPL ≈ 500). If the reviewer could provide more concrete references or implementation details for this attack, we would be happy to conduct additional experiments following those specifications.
> >
> > **Inversion attack:** We evaluate a plausible inversion attack based on the intuition that an adversary might try to reconstruct the selector matrix S and then reverse-engineer each cluster’s green list. In practice, this attack is particularly challenging because key details required to construct the selector matrix, such as the number of clusters L, the k-means initialization seed, and the dataset used to extract hidden states, are kept private by the model provider. Even if the attacker guesses L, reproducing the original hidden-state–to–cluster assignments is highly sensitive to both the k-means seed and the underlying dataset. To demonstrate this, we construct selector matrices using different datasets (OpenWebText vs. FineWeb) and different seeds, then measure clustering agreement on a shared test set using Adjusted Rand Index (ARI) and Normalized Mutual Information (NMI). The average scores (ARI ≈ 0.43, NMI ≈ 0.65) show disagreement, indicating that the true clustering cannot be reliably recovered. This suggests our method is robust to such inversion attacks. If the reviewer has other specific extraction strategies in mind, we would be happy to evaluate them and include the results in the revision.
> >
> > The details and results of all our new experiments are provided in Appendx K.
> >
> > ---
> >
> > We hope this response has addressed your concerns and would request you to kindly reconsider your overall assessment. Please let us know if we can address any further concerns.
> >
> >
> > [1] Gloaguen, T. et al. (2025). **[Towards Watermarking of Open-Source LLMs](https://arxiv.org/abs/2502.10525)**.
> >
> > [2] Penedo, G. et al. (2024). **[The FineWeb Datasets: Decanting the Web for the Finest Text Data at Scale](https://arxiv.org/abs/2406.17557).**

---

### Author Response · Authors · 2025-11-24

Dear Reviewers,

Thank you for your valuable feedback on our submission. We have now submitted a revised manuscript, with all changes made in response to your comments highlighted in **blue**. We hope our revisions and responses address your concerns.
If anything remains unclear or if further experiments or clarifications would be helpful, please feel free to let us know. We are happy to engage further and improve the paper.

---

### Meta-Review · Area_Chair_Jag1 · 2025-12-17

**Summary:**

The paper is on the borderline. There are two reviewers who vote for rejection in the first round of review. Among the concerns of these two reviewers, AC is not convinced by the rebuttal on the concerns regarding 1) novelty and 2) paraphrasing robustness. There is no clear evidence that both reviewers will change their score from negative to positive. Therefore, AC would recommend reject.

**Reviewer Concerns:**

Reviewer concerns AC believes are still outstanding:

1. Incremental novelty.

AC's comment: AC believes the current rebuttal about the novelty might be hard to convince the reviewer, as it is very short and no concrete evidence is provided. AC believes most likely, the reviewer will not change his/her score.

2. Paraphrasing robustness.

AC's comment: The authors admitted that this is a limitation of this paper.

AC is comfortable with the rebuttal to other concerns. In particular:

1. Attack model coverage.

AC's comment: This is a common concern from multiple reviewers. The authors add supplemental experiments which help alleviate the concern.

2. Model scale generalizability.

AC's comment: The authors provide extra experimental results on more models.

3. Limited variability.

AC's comment: An extra experiment was provided in the rebuttal.

**Reviewer Scores:**

The paper is on the borderline. However, there is no clear evidence that the two reviewers with a score of 4 are willing to change their evaluation from negative to positive. AC conjectures that two concerns (see above) from Reviewer 2c5R were not fully addressed by the rebuttal. Therefore, he/she might not change the opinion.

---

### Decision · Program_Chairs · 2026-01-26

Reject